# Active Optics—Progress in Modeling of Tulip-like Variable Curvature Mirrors

**Gerard R. Lemaitre** [1,2,*] **, Pascal Vola** [1,3] **and Patrick Lanzoni** [1,3]

1    Laboratoire d'Astrophysique de Marseille—LAM, 38 Rue Frédéric Joliot-Curie, 13388 Marseille, France
2    Campus Pharo, Aix Marseille Université—AMU, 13007 Marseille, France
3    Centre National de la Recherche Scientifique—CNRS, 13009 Marseille, France
*    Correspondence: gerard.lemaitre@lam.fr

**Abstract:** We present new results obtained from the modeling of a *tulip-like* variable curvature mirror (VCM) in the case of a central force that reacts to its contour. From Nastran finite element analysis, we shows that 3-D optimizations, using *non-linear static flexural option*, with an appropriate solution sequence, provide an accurate *tulip-like* VCM thickness distribution. This allows us to take into account boundary conditions, including the thin outer collarette and its link to a rigid ring. Modeling with a quenched stainless steel chromium substrate provides diffraction-limited optical surfaces. Rayleigh's quarter-wave criterion is performed over a *zoom range* from flat up to $f/3.5$ convexity over a 13 mm clear aperture and 10 daN central force. The optical testing results of a prototype *tulip-like* VCM elaborated from the previous analytic theory, show quasi-diffraction-limited figures for a zoom range up to $f/5$. The present modeling results should significantly help in the future construction of such VCMs with a zoom range extended up to $f/3.5$.

**Keywords:** active optics; modeling; variable curvature mirrors; zoom mirrors; finite element analysis; elasticity theory; actuator





## 1. Introduction

Geometrical deformable configurations with the ability to generate *variable curvature mirrors* (VCMs) were discovered by Lemaitre [1,2]. These three configurations were derived from the elasticity theory of the small deformations of thin plates with variable thickness distributions. Such VCMs—sometimes called *zoom mirrors*—have either a cycloid-like or tulip-like thickness distribution.

The first manifold is a *cycloid-like form* that requires a uniform load—as air pressure—applied over the mirror back surface and reacts in a ring force along its circular contour without requiring any bending moment. This particular VCM configuration is of practical interest because it can easily generate accurate optical curvatures varying from *plane at rest* up to $f/3.5$ when *under stress* [2].

The second manifold are *tulip-like forms* where the static equilibrium is a combination of two sets of acting forces: a central axial force in reaction to either (i) a uniform load or (ii) an axial perimeter ring force. We present here the first case of a *tulip-like* VCM that provides, from modeling, a zoom range from *plane at rest* up to $f/3.5$ when *under stress*.

For such a large zoom range, Ferrari and Lemaitre [3] and Ferrari [4] carried out analytic investigations with the elasticity theory of large deformations. This theory takes into account *radial and tangential* stresses at the middle surface of the plate. A review of these theories and construction results can be found in *Astronomical Optics and Elasticity Theory—Active Optics Methods* by Lemaitre [2]. Some preliminary prototype *tulip-like* VCMs were built around the 1990's at the Observatoire de Marseille—moved to the newly created Laboratoire d'Astrophysique de Marseille (LAM) in the 2000s—that were equipped with a motorized force actuator. A variable curvature mirror, generated by perimeter bending moments, was developed in a search as an optimal variant for a resonator of an electric

discharge $CO_2$ laser. This was studied using either a germanium exit mirror or metallized mirrors with central apertures of different diameters (Belomestnov et al. [5]). Grazing incidence mirrors were actively bent into an ellipsoidal shape to increase the detection efficiency. This was obtained by applying bending moments at their contour as developed for micro focalization experiments at the European Synchrotron Radiation Facility (ESRF) (Ziegler et al. [6]). Characteristics of a mechanically bent-shaped mirror for X-ray optics using a long trace profiler (LTP) was evaluated by Kamachi et al. [7], where the "*arm method*" mirror bender controls the mirror curvature with little effect on the slope error. Modeling of a VCM designed as a thin elastic plate with an exponential thickness distribution was actuated with a uniform pressure under simply supported boundary conditions (Xie et al. [8]). An active optics method, for making smooth and curved freeform mirrors from an elastically deformable matrix, was developed to generate curvature modes using bending moments. The matrix replication technique provided the curved astigmatism correction of a reflective off-axis segment of a Schmidt plate without the need to make a full-size plate (Lemaitre and Lanzoni [9]).

The remainder of the paper is organized as follows: In Section 2, we introduce the theory. Section 3 describes the introducing features: optical focal-ratio, buckling instability, VCM zoom range and metal choice. Section 4 is the simulation methodology. Section 5 describes the simulation results. Section 6 details the experimental results. Finally, Section 7 presents the conclusions.

## 2. Theory—Thin Circular Plate VCMs

### 2.1. Preliminarily Definition of the Curvature Mode

The first-order modes of the triangle optical matrix characterizing a wavefront shape are the curvature mode (*Cv*-1) and tilt mode (*Tilt*-1). These are the two fundamental modes involved in Gaussian optics. As a tilt mode is trivially obtained by a global rotation of a rigid substrate, investigations to achieve elastic deformation modes only reduce to deformable mirrors generating the *Cv*-1 curvature mode.

Let us denote $z(r)$ the figure achieved by the flexure of a circular plate, which is flat at rest. In the *thin plate theory of small deformations*, the *Cv*-1 curvature mode is represented by a paraboloid flexure:

$$z = A_{20}r^2 \equiv \frac{1}{2R}r^2 \tag{1}$$

where $A_{20}$ is a constant coefficient of the optics triangle matrix [2], and $1/R$ the variable optical surface curvature.

The two classes of thickness that can provide a curvature mode *Cv*-1 with the thin plate theory are *constant thickness distributions* (CTDs) and *variable thickness distributions* (VTDs) [2]. The first class requires a larger aperture diameter than that required, whilst the second class is the easiest for practical reasons.

### 2.2. Analytic Theory—VCM with Constant Thickness Distribution (CTD)

*Constant thickness distributions* (CTDs) may generate the first-order curvature mode *Cv*-1 provided a constant bending moment is applied along the circular contour of the plate. This requires VCM designs that use concentric two-zone configurations and opposite loading circular forces in each area [2]. We can remark that CTDs provide a conveniently accurate *Cv*-1 mode only for the inner zone.

### 2.3. Analytic Theory—VCMs with Variable Thickness Distribution (VTD)

Let us consider a plane circular plate with variable thickness $t(r)$ having a rigidity $D(r)$ classically expressed by:

$$D(r) = Et^3(r) / \left[ 12\left(1 - v^2\right) \right] \tag{2}$$

where $E$ and $\nu$ are the Young's modulus and Poisson's ratio, respectively. Denoting $a$ and $t_0$ the edge radius and a dimensional mean thickness, respectively, the non-dimensional parameters $\rho$ and $T_{20}$ can be defined as:

$$\rho = r/a \quad and \quad T_{20} = t/t_0 \tag{3}$$

The thin plate theory applied to VCMs leads to three manifolds of *variable thickness distributions* (VTDs) as elaborated for active optics applications. The first-order curvature mode, *Cv*-1, is achieved by the use of easy load systems where the thicknesses $t(r)$ are denoted *cycloid-like* forms or *tulip-like* forms (Figure 1).

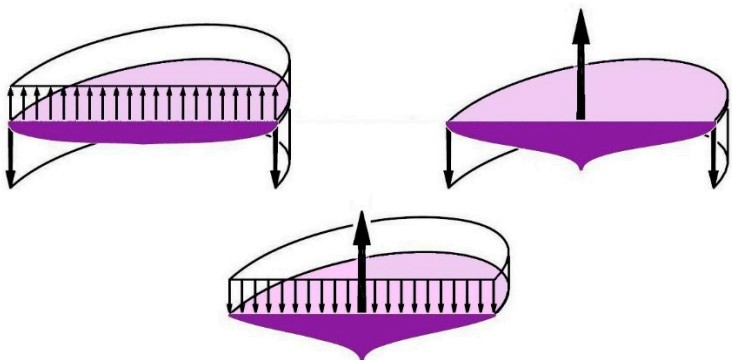

**Figure 1.** VCMs derived from the VTD class leading to a *cycloid-like* form and two *tulip-like* forms. The three dimensionless thicknesses $T_{20}$, with $\rho = r/a$, $\rho \in [0, 1]$ (Lemaitre [2]), are as follows: *top left:* cycloid-like VCM with uniform loading and reaction at the edge, $T_{20} = (1 - \rho^2)^{1/3}$; *top right*: tulip-like VCM with axial force at center and edge reaction, $T_{20} = (-\ln \rho^2)^{1/3}$; *down*: tulip-like VCM with uniform loading and central reaction, $T_{20} = (\rho^2 - \ln \rho^2 - 1)^{1/3}$.

As it is easily achievable in practice by means of a single force actuator and without requiring any uniform load, we consider, hereafter, only the case of a *tulip-like* VCM with a central force and edge reaction. The analytic dimensionless thickness—before the finite element analysis—is then $T_{20} = t/t_0 = (-\ln \rho^2)^{1/3}$, for which we defines preliminarily associated quantities as the central force acting on the VCM hereafter.

### 2.4. Tulip-like VCM with a Central Force and Edge Reaction

From the definitions of radial and tangential bending moments, $M_r$ and $M_t$, respectively, the axisymmetric case of a thin circular plate leads to [2,10]:

$$M_r = D\left(\frac{d^2z}{dr^2} + \frac{\nu}{r}\frac{dz}{dr}\right) \quad and \quad M_t = D\left(\frac{1}{r}\frac{dz}{dr} + \nu\frac{d^2z}{dr^2}\right) \tag{4}$$

and denoting $Q_r$ as the shearing force, the static equilibrium of a plate element can be written as:

$$M_r + r\frac{dM_r}{dr} - M_t + rQ_r = 0 \tag{5}$$

After substitution, the resulting equation is:

$$D\frac{d}{dr}\left(\nabla^2 z\right) + \left(\frac{d^2z}{dr^2} + \frac{\nu}{r}\frac{dz}{dr}\right)\frac{dD}{dr} = -Q_r \tag{6}$$

where the Laplacian value is $\nabla^2 z = 4A_{20}$ from substitution in Equation (1). The shearing force $Q_r$ depends on each case of the three VTDs associated to external loads that generate the curvature mode *Cv*-1 (Figure 1). Each of them requires a particular shearing force.

Recall the results obtained from the thin plate theory for a substrate deflected by an axial force $F$ applied to its center that gives rise to a uniform reaction—$F$—at the edge contour. If we consider an equivalent uniform load $q$ applied to its entire surface, we may define the equivalent central force by $F = \pi a^2 q$ applied onto a tiny area. The associated shearing force $Qr$ satisfies $F + 2\pi r\, Qr = 0$. We also select a null bending moment at the edge $r = a$ by selecting the rigidity $D(a) = 0$. Integration lead us to the rigidity [2]:

$$D = -\frac{FR}{4\pi(1+\nu)}\left(-\ln\frac{r^2}{a^2}\right) \tag{7}$$

As the infinite thickness at $r = 0$ and of the vertical tangents at the substrate edge, we call this thickness distribution a *tulip-like form*.

**Theorem 1.** *From the analytic elasticity theory of thin plates, a variable curvature mirror of curvature 1/R is obtained by an axial force F at the center and a reaction at the edge if, and only if, its thickness distribution $t = T_{20}\, t_0$ is a tulip-like shape, such as:*

$$T_{20} = \left(-\ln\frac{r^2}{a^2}\right)^{1/3} \quad with \quad \frac{t_0}{a} = -\left[3(1-\nu)\frac{FR}{\pi E a^3}\right]^{1/3} \tag{8}$$

*where the product FR is negative.*

The thickness distribution is in the form $(-\ln\rho^2)^{1/3}$ around the center and then $T_{20}(0) \to \infty$. As this distribution looks like a stem at the center, it has been called a *tulip-like* VCM. In practical applications, it is always possible to limit the central thickness to a finite value. A convenient truncation of the stem allows respecting optics Rayleigh's quarter-wave criterion ($\lambda/4$) during the flexure around the central area. The axial force $F$ is then applied onto a small area, say, typically at a stem radius of $a/50$.

## 3. Optical Focal Ratio, Buckling Instability, VCM Zoom Range and Metal Choice

### 3.1. Optical f-Ratio

We can determine an optical $f$-ratio generated by the $Cv$-1 deformation mode. Assuming a flat mirror when in an unstressed state, let us define this $f$-ratio as $\Omega$. Considering a mirror diameter $2a$,

$$\Omega = |f/2a| = |R/4a| = |1/(8a\,A_{20})|. \tag{9}$$

Recalling that the central force $F$ is related to a uniform load $q$ by $F = \pi a^2 q$ and that $t_0 = t/T_{20}$, we obtains, as a function of the $f$-ratio $\Omega$,

$$\frac{t}{a} = \left[12(1-\nu)\Omega\frac{F}{\pi E a^2}\right]^{1/3} \tag{10}$$

### 3.2. Buckling Instability

*Self-buckling instability* may happen during a curvature change. This effect is similar to the meniscus shell "jumping toy" in polymer material, which is manually brought, temporarily, to opposite curvature. Avoiding buckling instability requires taking into account the radial and tangential tensions $N_r$, and $N_t$, which exist at the middle surface and showing that the maximum compression value of $N_r$ shall remain small compared to a critical value.

Self-buckling instability is avoided by restricting curvatures to always having the same algebraic sign during *zooming*. Furthermore, all three VTDs decreasing to zero towards the edge prevents this instability.

### 3.3. VCM Zoom Range

As self-buckling instability is avoided, it is useful to consider a basic alternative where a VCM is polished flat when unstressed, and define a *zoom range* as follow: All curvatures $1/R_i$ should remain in the same algebraic sign, i.e., remaining in the same direction of the curvature when stressed [2]. To prevent self-buckling instability, this lead us the anti-buckling criterion as follows:

*All varying curvatures $1/R_i$ shall remain in the same direction for the whole zoom range.*

From various *cycloid-like* VCM experiments in quenched stainless steel alloy Fe67Cr13, with aspect ratios $2a/t(0) = 50$ and zoom ranges varying in convexity from $f/\infty$ to $f/3.5$, it has been shown that the middle-surface curvature will not entail any self-buckling effect.

### 3.4. Metal Choice

We select VCM substrates in *quenched chromium stainless steel* alloy Fe67Cr13 with *post-quenched ageing* from Ugitex Corp. [11] which shows a large *elastic deformability* much superior to that of fused silica or glass ceramics. Deformability is characterized by the ratio of maximum working stress over the Young's modulus, i.e., the $\sigma_{M.W.S}/E$ ratio.

Other stainless steel chromium alloys with somewhat larger elastic deformability than the quenched Fe67Cr13 may exist, as selected here, for instance, by including 1–2% *molybdenum*. Other linear alloys, i.e., showing linear stress–strain relationships, are of interest for VCMs, such as *titanium* alloy Ti90Al6V4 or *beryllium* alloy Be95Cu5, but are more brittle. However, minimizing the lathe-machining *chip-cutting size* should be tested for the lathe-cutting finishing operation.

## 4. Simulation Methodology—FEA of a Tulip-like VCM Bent by a Central Force

From Section 2, concerning the *small deformation theory*, the flexural results are valid only if the flexural sags $z_{max} = a^2/2R$ are small compared to the substrate mean thickness $t_0$, i.e., when the relationship of flexure vs. loading ($z_{max}$, $q$) is linear. When sags $z_{max}$ become non-negligible compared to the thickness $t_0$, the radial and tangential stresses at the plate middle surface must be considered. Investigations with the *large deformation theory* of thin plates led us to conclude that *non-linear* results cannot be taken into account for convenient boundaries at the contour. Due to the remarkable calculation accuracy of the MSC Nastran [12] code by the MSC software code of *finite element analysis* (FEA), we present, hereafter, results derived from Vola. This uses 3-D optimizations with the large *non-linear static flexural option* applied to a model where the VCM could be linked, in a single piece, to an outer rigid ring via a very thin *collarette* (Figure 2).

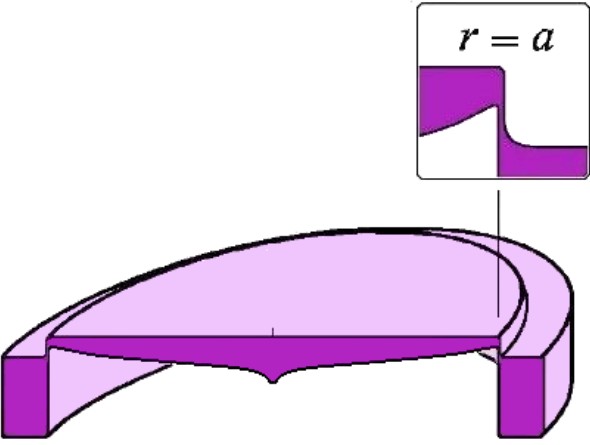

**Figure 2.** Schematic of a *tulip-like* VCM with central truncated stem showing the plate-collarette-ring geometry.

### 4.1. Modeling of VCM for the Plate Alone

Each finite element boundary provides equilibrium by use of a complete local equation set. FEA analyses was carried out preliminarily with VCM thickness $t = t_0 \, (-\ln \rho^2)^{1/3}$, $2a = 16$ mm diameter, in quenched stainless steel chromium FeCr13 alloy and *post-quench ageing* that shows high ultimate strength with Young's modulus $E = 205$ GPa and Poisson ratio $\nu = 0.315$.

The central pushing force, equivalent to $F \equiv \pi a^2 q$, is in an axial reaction with the mirror contour. Due to the *infinite thickness* at center from the thin plate theory, $T = t/t_0 = [-\ln(r^2/a^2)]^{1/3}$, the mirror design is conveniently truncated near the center. The stem truncation provides a negligible effect with respect to the quarter-wave Rayleigh's criterion.

Preliminarily, the geometry of the plate alone is defined by a truncated central thickness as $t_c = 618$ µm, expanding up to $r = 0.1$ mm. We define the equivalent thickness as $t_0 = 300$ µm for $r = 4.85$ mm. We also assume a conical shape for $7.6 < r < 8$ mm that leads to $t(a) = 100$ µm.

We consider Nastran modeling with the solution sequence SOL106 (Vola) where the central push forces from $F = 0$ to $F = 14$ daN generate a *convex* shape. The flexural deviation from a paraboloid shape is processed by the least mean square. The Nastran boundary conditions are as follows: all nodes at the edges of the VCM are fixed along the mirror axis (Z-axis) but are free to move radially. In addition, we remark that the model, in geometry and in loads, is axisymmetric. This means that no tangential displacement, at any node of the mesh, is permitted. To achieve this, we model only a quarter of the mirror and impose onto all nodes of the two created radial sections a null *out-of-plane* displacement; in other words, with degree-of-freedom (DOF) terminology, an X-DOF is fixed on section YZ, and Y-DOF is fixed on section ZX.

The results confirm that the deflections $z(r)$ are *non-linear* (Figure 3). Nastran shows deviations $\Delta z(r)$ from a purely paraboloid shape (Figure 4). The model gives a total sag of 377 µm for a medium intensity force of 10 daN (Figure 5).

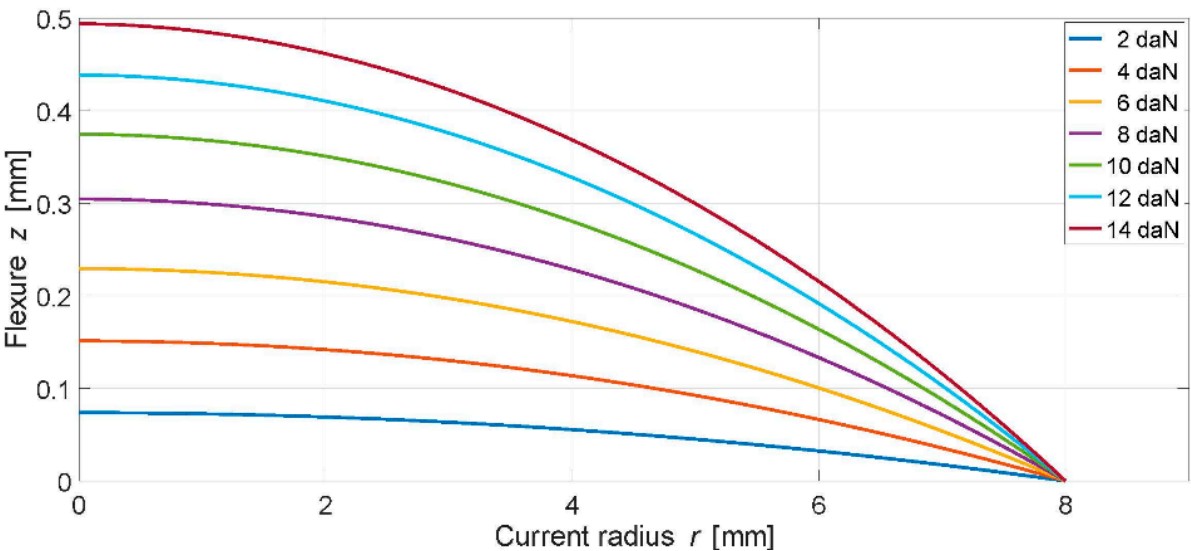

**Figure 3.** Flexures vs. current radius for *tulip-like* VCM. Thickness $T = t/t_0 = [-\ln(\rho)^2]^{1/3}$ for $0.0125 < \rho < 0.95$ and $t_0 = 300$ µm. Truncated central thickness $t_c = 618$ µm for $\rho < 0.0125$. Conical thickness for $\rho > 0.95$ with $t \, (\rho = 1) = 100$ µm.

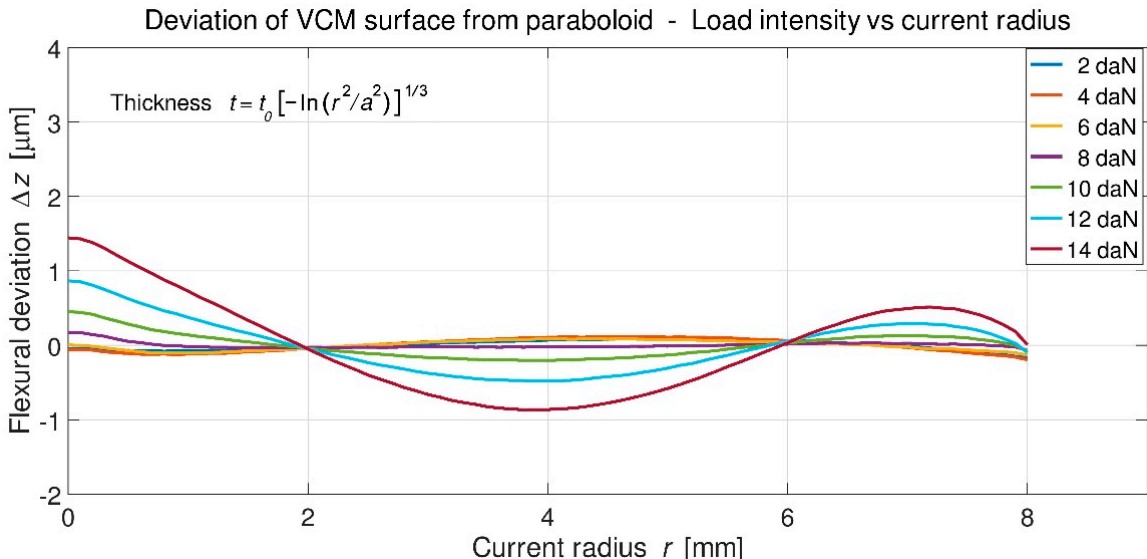

**Figure 4.** Least mean-square flexural deviations. Thickness distribution as before.

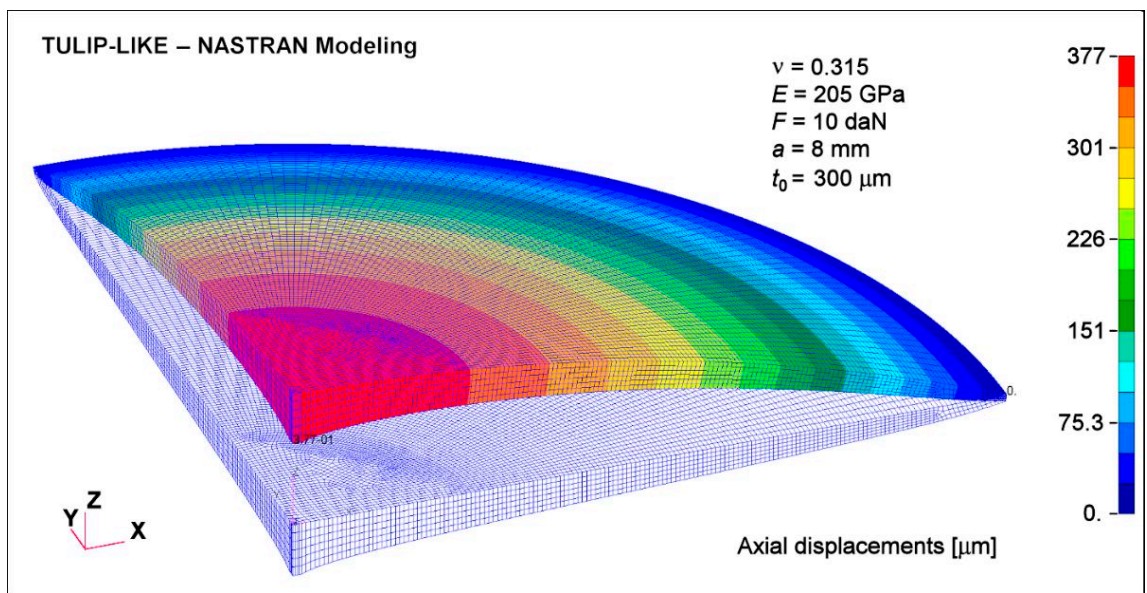

**Figure 5.** Axial displacements for medium intensity force *F* = 10 daN. Thickness distribution as before.

Due to an important local effect shown as a bump shape near the center by of the previous flexural deviations (Figure 4), one somewhat increases the central diameter of the stem to provide a better optimization with Nastran next calculation. Conserving the central thickness $t_c$ = 618 µm as unchanged, we expand the truncation radius up to 0.4 mm (instead of 0.1 mm), i.e., a truncation at $\rho$ = 0.05 (instead of 0.0125 previously). The flexural deviation is then improved (Figure 6).

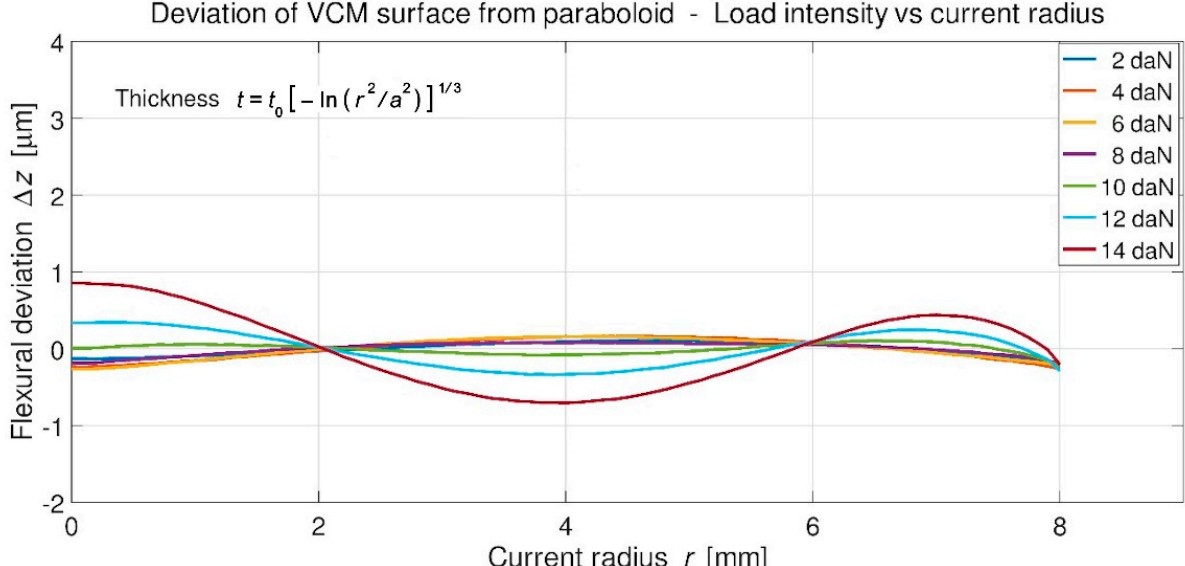

**Figure 6.** Least mean-square flexural deviations with truncated radius at $\rho = 0.05$ and stud thickness $t_c = 618$ μm. Thickness $T = t/t_0 = [-\ln(\rho^2)]^{1/3}$ for $0.05 < \rho < 0.95$ and $t_0 = 300$ μm. Conical for $\rho > 0.95$ with $t(1) = 100$ μm.

### 4.2. Modeling of VCM with an Outer Cylinder Collarette and Rigid Ring

For the practicable reasons of fabrication and avoiding interface problems, the *tulip-like form* VCM plate shall be completed in a monolithic geometry using the implementation of an outer radially thin cylinder—or collarette—linked to a rigid outer ring. The combination plate-collarette-ring is a one-piece construction. Considering the flexural deviations obtained above that show the lack of rigidity of the plate conical area, we slightly enlarge the conical area and somewhat increase its edge thickness.

The Nastran model is built with the same logic as the previous one without the collarette. The degree-of-freedom Z-DOF boundary condition is now applied on the base of the rigid ring instead of on the mirror edge. Here, we also use the axisymmetry of the problem for modeling only a quarter VCM and then fixing X-DOF on the radial section YZ and Y-DOF on the radial section ZX. The VCM edge thickness is now $t$ ($\rho = 1$) = 120 μm, and the narrow collarette is set up with three-layer meshing for a total thickness of 20 μm (Figure 7). The collarette flexibility ratio over plate-edge thickness is $1/6^3 = 1/216$, which provides a negligible bending moment at the contour, i.e., an idealized articulation.

Flexural deviations are carried out with the central truncation at $\rho = 0.05$. The central thickness $t_c = 618$ μm is unchanged and looks like a tiny stud at the center. The results show important deviations in this case (Figure 8).

Due the important flexural deviation obtained with this complete geometry, we restrain, hereafter, (i) in limiting the maximum central force up to $F = 12$ daN and (ii) in optimizing the flexural deviations to a current radius $r \leq 7$ mm, i.e., $\rho \leq 0.875$. Furthermore, we will investigate the modifications for two cases of thickness distribution in the VCM area for $0.05 < \rho < 0.9$.

In order to reduce the amplitude of the flexural deviations, we now consider a thickness $t/t_0 = [-\ln \rho^2 - \varepsilon\,(\rho^2 - 2\rho^4 + \rho^6)]^{1/3}$, where $\varepsilon$ is a dimensionless parameter. The optimization process with $\varepsilon = 0.6$ leads us to a maximum flexural deviation max($\Delta z$) = 1.4 μm up to $F \leq 12$ daN.

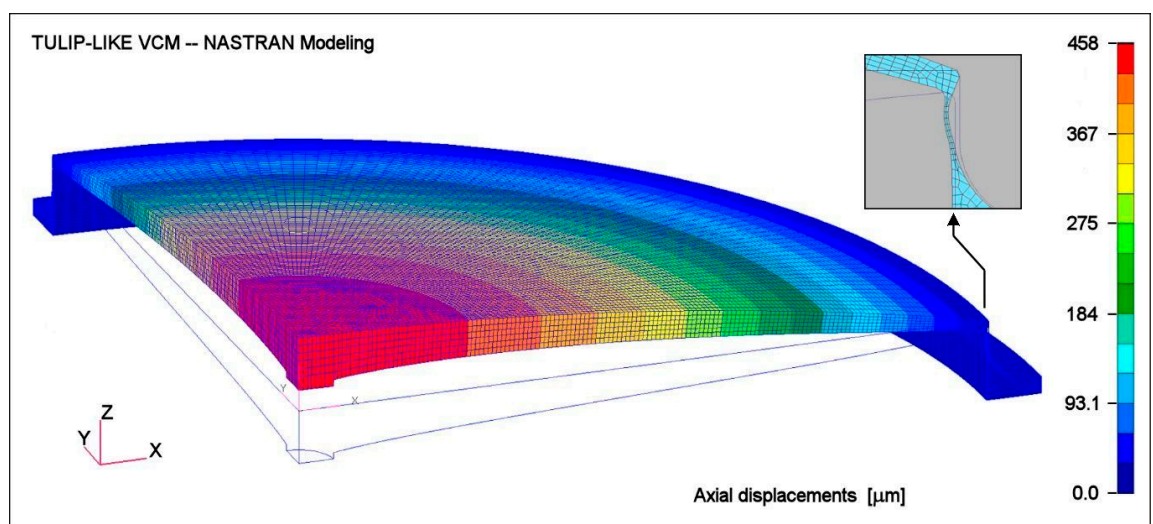

**Figure 7.** Axial displacements for maximum force $F$ = 14 daN. Mesh model plate-collarette-ring. $T = t/t_0 = (-\ln \rho^2)^{1/3}$ for $0.05 < \rho < 0.9$ and $t_0 = 300$ μm. A conical shape is designed for $0.9 < \rho < 1$ with $t (\rho = 1) = 120$ μm. Collarette thickness 20 μm.

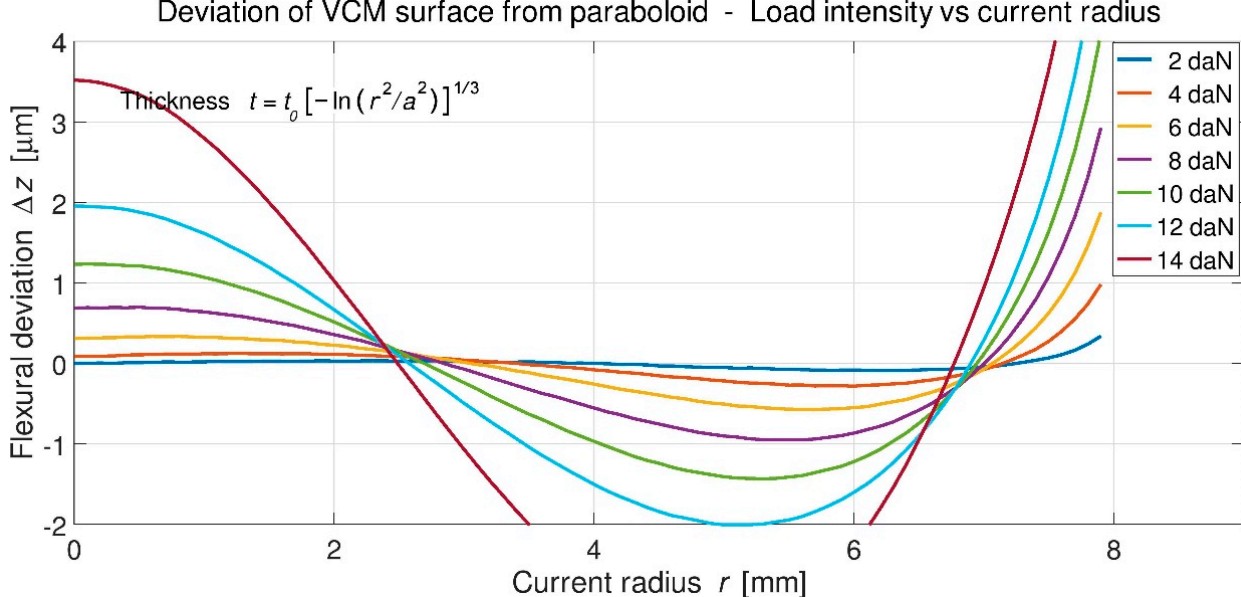

**Figure 8.** Least mean-square flexural deviations. Conical shape for $0.90 < \rho < 1$. $t(1) = 120$ μm. Collarette 20 μm width.

The second investigation, made with $t/t_0 = [-\ln \rho^2 - \varepsilon\,(\rho^4 - \rho^6)]^{1/3}$ and $\varepsilon = 0.4$, provides a similar maximum deviation $\max(\Delta z) = 1.5$ μm up to $F \leq 12$ daN. However, for the range $F \leq 10$ daN, the flexural deviation is the best one with $\max(\Delta z) = 0.7$ μm, so we select the latter result (Figure 9). The total deflection amount vs. forces for the full range 2–12 daN is updated (Figure 10). The final resulting geometry $t(r)$ from the FEA design (Table 1) includes the thickness $t^*\,(r)$ for the CNC lathe machining with 10 μm constant over-thickness. A scale drawing of the front part of the VCM, as investigated below for the FEA analyses, includes the degree-of-freedom (DOF) of the boundary conditions of the outer ring (Figure 11).

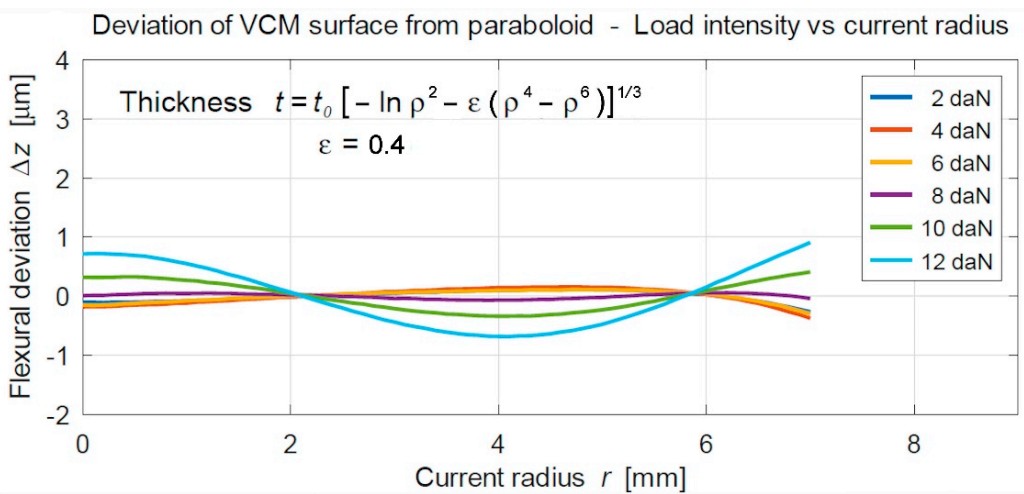

**Figure 9.** Flexural deviations from paraboloid for central forces up to *F* =12 daN over a 14 mm aperture diameter. In the range *F* ≤ 10 daN over 13 mm aperture diameter, deviations to a paraboloid remain smaller than 0.6 μm PTV, i.e., 0.1 μm RMS.

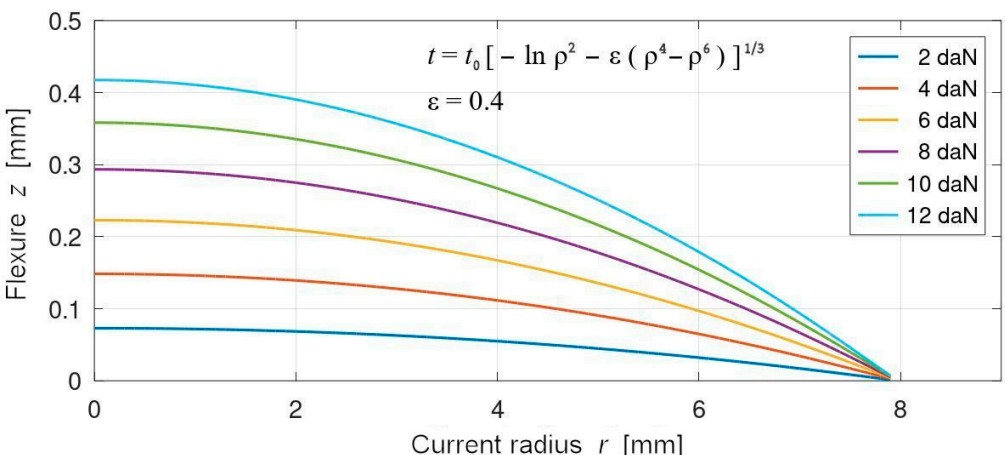

**Figure 10.** Flexure amplitude vs. current radius for central forces in the range intensity 2–12 daN over 16 mm full diameter. The flexural sag is 418 μm for *F* = 12 daN and 355 μm for *F* = 10 daN.

**Table 1.** Thickness *t*(*r*) of *tulip-like* VCM from Nastran modeling. Truncated center thickness $t_c$ = 618 μm for *r* < 0.4 mm. $t_0$ = 300 μm. Radial thickness of outer cylinder collarette 20 μm. Thickness *t*\*(*r*) is *t*(*r*) + 10 μm constant over-thickness for CNC lathe machining.

| $t/t_0 = [-\ln \rho^2 - \varepsilon (\rho^4 - \rho^6)]^{1/3}$ for 0.4 < *r* = ρ, *a* < 7.2 mm. Conical 7.2 < *r* < 8, ε = 0.4 | | | | | | | | | |
|---|---|---|---|---|---|---|---|---|---|
| Units: *r* [mm] *t*, *t*\* [μm] | | | | | | | | | |
| *R* | 0 | 0.4⁻ | 0.4⁺ | 4 | 6 | 7.2 | 8⁻ | 8 | 8⁺ | 12 |
| *t* | 618 | 618 | 545 | 333 | 241 | 163 | 120 | 6750 | 6000 | 6000 |
| *t*\* | 628 | 628 | 555 | 343 | 251 | 173 | 130 | 6750 | 6000 | 6000 |

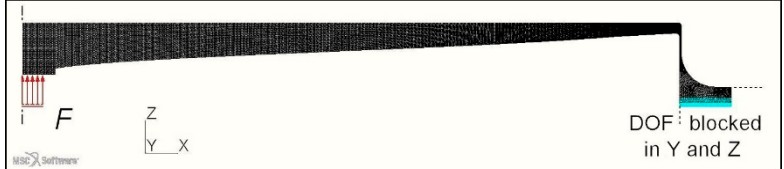

**Figure 11.** Drawing section, in $(x, z)$ plane, of the VCM front part showing the degree-of-freedom (DOF) boundaries. These conditions applies to the previous FEA modeling as *blocked*, i.e., non-movable, in the axial and tangential directions, i.e., $z$ and $y$ and is free to move in the $x$-direction. The force $F = 10$ daN is applied over $\rho \leq 0.03125$, i.e., 0.25 mm radius area.

The definition layout for the lathe-turning execution (Figure 12) requires including some extra-thicknesses before cubic boron nitride (CBN) rectification of the collarette outer part at 18,000 rpm, where its 50 µm radial thickness becomes 20 µm. The next operations involve figuring the VCM optical surface with 6 and 3 µm diamond grain-size and polishing with $Al_2O_3$ of 1, 0.3 and 0.05 µm grain-size on pitch tools.

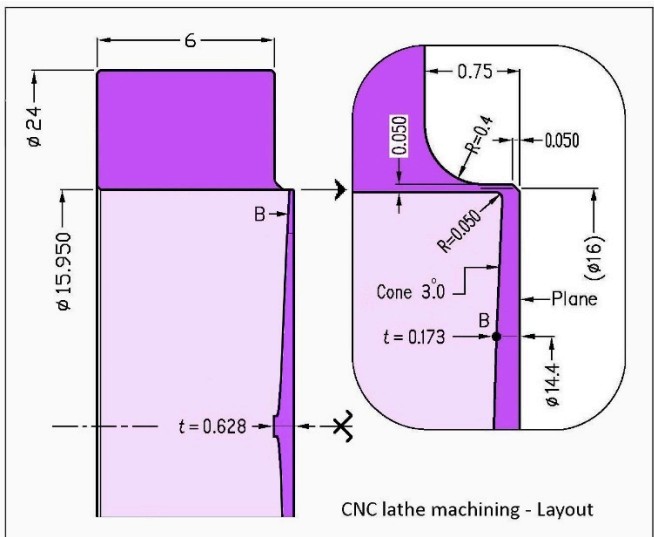

**Figure 12.** Definition layout for the CNC lathe-turning execution. Dimension units (mm).

## 5. Simulation Results

### 5.1. Final Data of the Tulip-like VCM Modeling

We summarize the modeling parameters of the VCM as designed from Table 1 and Figures 11 and 12. The VCM is in *quenched chromium stainless steel* Fe87Cr13 alloy with *post-quench ageing*. For practical applications, we take into account a restrain force range up to $F \leq 10$ daN corresponding to a 355 µm flexural sag over 16 mm diameter. This later force range provides diffraction-limited optical surfaces for a clear aperture diameter of 13 mm. The elasticity data and modeling optical geometry are shown in Table 2.

**Table 2.** Results from optics and elasticity parameters of the *tulip-like* VCM modeling.

| | | | |
|---|---|---|---|
| Poisson ratio | $\nu = 0.315$ | Young's modulus | $E = 205$ GPa |
| Mean thickness | $t_0 = 300$ µm | Nastran optimal profile | $0.4 < r < 7.2$ mm |
| Central cut thickness | $t_c = 618$ µm | Collarette radial thickness | $\Delta r = 20$ µm |
| Force range | $F \leq 12$ daN | Collarette stress max | $\sigma_{max} = 920$ MPa |
| Flexural sag $F = 10$ daN | $z_0 = 355$ µm | Radius of curvature | $R = 90.1$ mm |
| Outer diameter | $2a = 16$ mm | Zoom $f$-ratio | $f/\infty$–$f/2.82$ |
| Clear aperture dia. | $d_{Opt} = 13$ mm | Zoom $f$-ratio | $f/\infty$–$f/3.47$ |

*5.2. Maximum Stresses, Pre-Stressing and Creep Deformation*

The Nastran results show that, for a central force $F$ = 10.1 daN, the stresses are not maximal at the surface of the plate but are, by far, along the collarette surfaces. The maximum tensile stresses of the Fe67Cr13 alloy, also named X30C13 by Aubert and Duval Corp. [13] and as denoted *ultimate stress*, is $\sigma_{ult}$ = 1700 MPa, and the corresponding 0.2% elastic elongation stress is $\sigma_{0.2}$ = 1500 MPa.

The previous 3-D mesh is not refined enough to allow a correct stress analysis of the collarette area. We use an axisymmetric calculation in which only the radial cross section of the VCM is modeled. The collarette is meshed with 12 layers of triangular elements. This allows the accurate determination of the Von Mises's stress areas. For a central force $F$ = 10.1 daN, the results from the stress analysis show a maximum stress of $\sigma_{max}$ = 2100 MPa at the collarette inner surface. Due to the convexity during loading, this stress corresponds to an elongation, which then overpasses the $\sigma_{ult}$ of the material. In fact, there is a metal *creeping—creep* deformation or *fluage* in French—of the metal that reduces the maximum stresses [14]. This creeping effect can be drastically modelled using Nastran by artificially introducing the lower Young's modulus $E$ = 100 GPa (instead of $E$ = 205 GPa) for the most stressed areas. This shows that the VCM can provide a much larger flexure than the results obtained from the basic calculation (Figure 13).

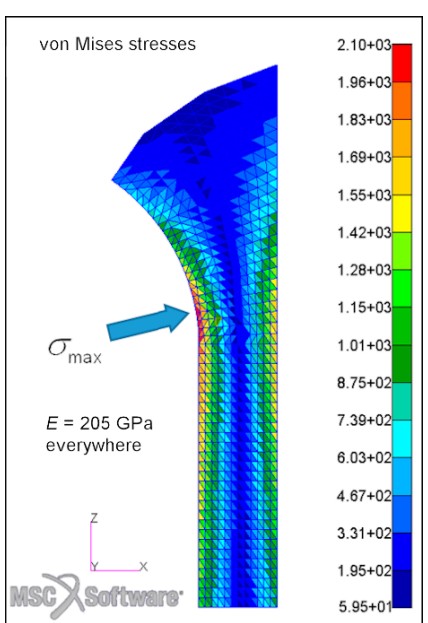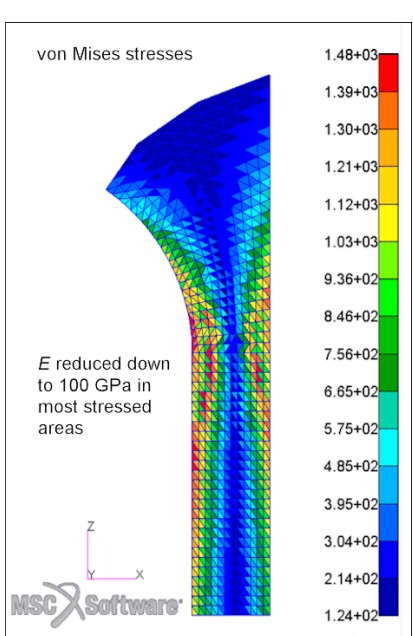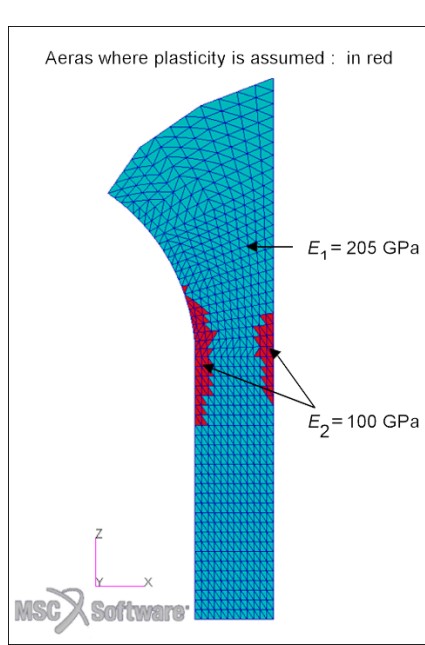

**Figure 13.** Results from stress analysis of the collarette at $F$ = 10 daN with a mesh axisymmetric model. For a constant modulus $E$ = 205 GPa everywhere, the scale shows a maximum stress $\sigma_{max}$ = 2100 MPa (*left*). Due to the overpassed limit stress of the material compared to $\sigma_{ult}$, we take into account the creeping effect—or *fluage*—by artificially introducing a smaller modulus $E$ = 100 GPa in the most stressed zones. We obtain, from Nastran, a more realistic stress distribution, where $\sigma_{max} \leq$ 1480 MPa (*center* and *right*).

## 6. Experimental Results

*6.1. Mechanical Assembly of a Prototype and Actuator*

The *tulip-like* VCM design uses a motorized translator lead screw as a force actuator that generates the curvature variation. A central thrust ball bearing provides contact to generate the axial push forces at the stem of VCM rear surface. This requires a preliminary set up prepositioning of the VCM and back-part assembly with a radial screw (Figure 14).

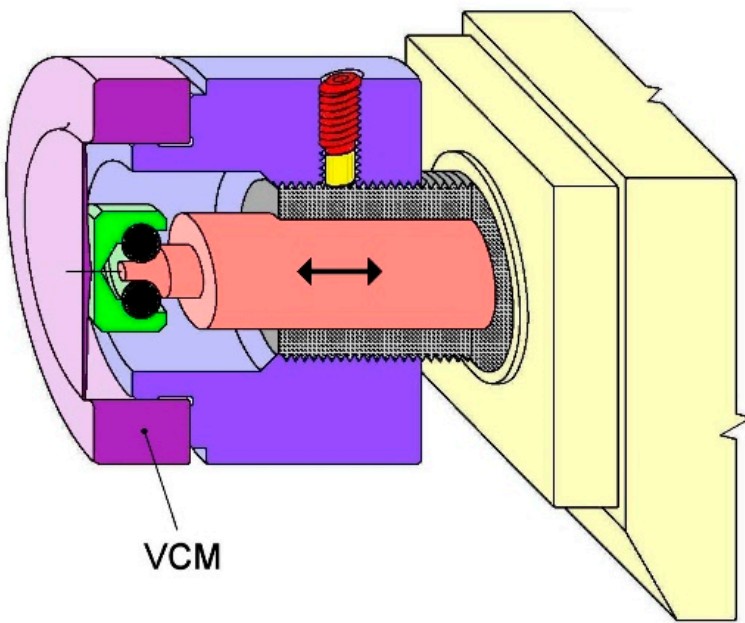

**Figure 14.** Tulip-like VCM assembly.

The VCM actuator is a high-precision motorized actuator made in France by Micro-Controle-Spectra Physics Corp. [15] and is able to deliver push forces up to 12 daN with a bidirectional repeatability of $\pm 0.30$ μm over a 25 mm travel range. The motor type is a UE17CC, 12V-DC, 0.2 Amp. Mazzanti elaborates specific codes with the implementation of a dedicated electronic PC card for control command. The translation driver module uses an encoder close-loop control with initialization set up and a limit switch for maximum VCM deflection [16] (Figure 15).

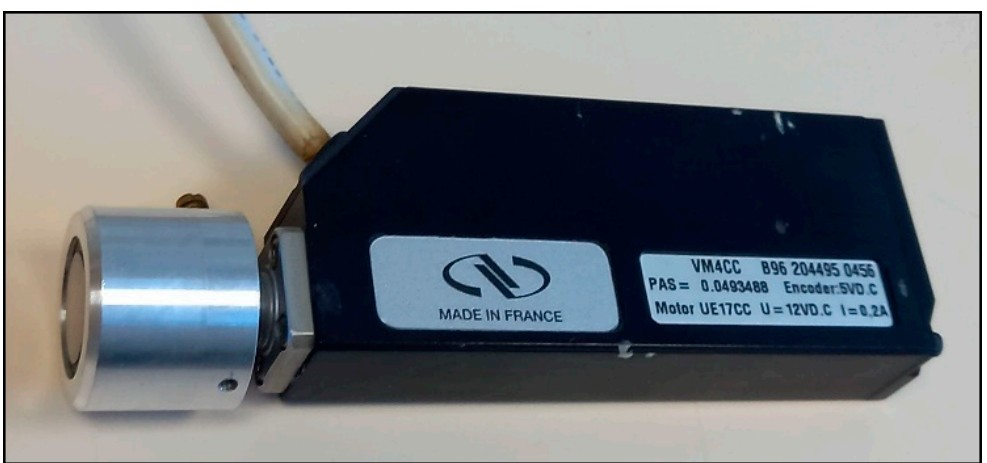

**Figure 15.** Actuator of the push force control system by Micro-Contrôle Corp.

### 6.2. Results from Realization and Optical Testing

Before the FEA optimization modeling presented in Section 4, a *tulip-like* VCM was designed from the analytic plate deflection theory, plate alone, i.e., without taking into account the effect of the outer collarette and ring. Some VCMs—a plate-collarette-ring in a single piece—were built by Gauthier Precisions Corp., using a NC lathe and polished at LAM by Lanzoni. A prototype VCM (Figure 16) was adapted to a motorized push force actuator. For a moderate zoom range from $f/100$ to $f/5$, the He-Ne interferograms with respect to curvature radii [2800, 420, 230 and 140 mm] show slight wave-front deviations

to paraboloid for the clear aperture 2*a* = 14 mm, then showing quasi-diffraction-limited results of the mirror surface (Figure 17).

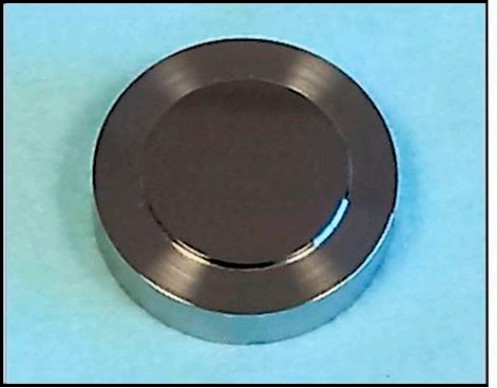 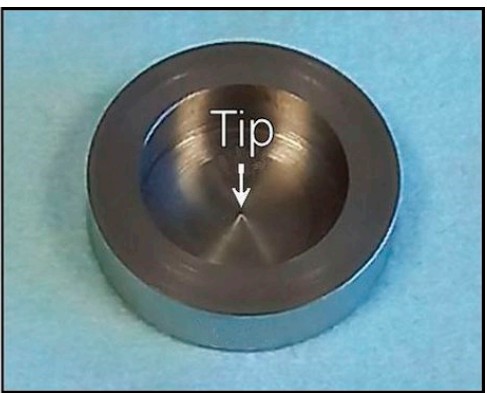

**Figure 16.** View of prototype *tulip-like* VCM#5 with its rear central tip.

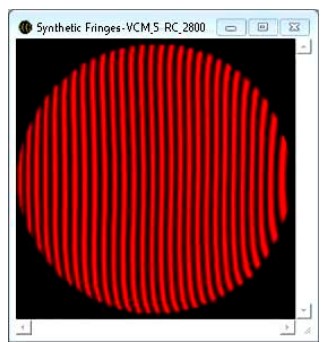 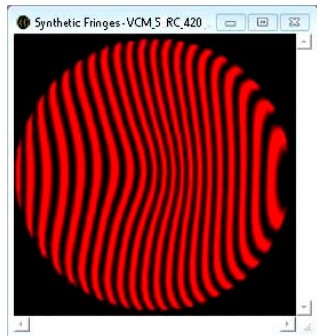 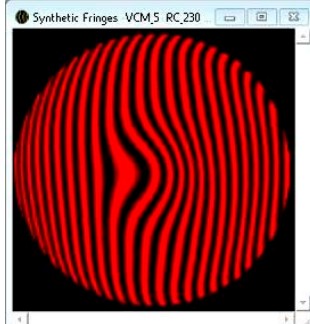 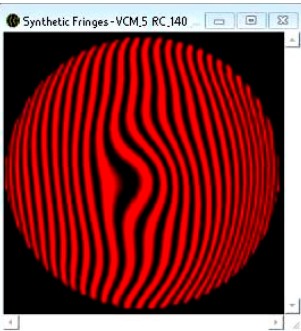

**Figure 17.** He-Ne interferograms of *tulip-like* VCM#5 without over-thickness at the central tip. Curvature variation from *f*/100, *f*/15, *f*/8.2 to *f*/5 (**left** to **right**) for a 14 mm clear aperture diameter. Curvature radii RC = 2800, 420, 230 and 140 mm, respectively. A realization by use of modeling from Table 1, i.e., including an over-thickness at the tip–should avoid present local deviation errors near the center.

The surface plot and aberration table of VCM#5 (Figure 18) are for the maximum central force corresponding to *f*/5 at RdC = 140 mm. It was designed with a thickness close to $T = t/t_0 = [-\ln(\rho^2)]^{1/3}$ as a preliminary shape, i.e., as in Figure 6, without over-thickness at the tip. This partly explains the central deviation error near the VCM center that would be improved by the use of the design in Table 1.

It may be remarked from Figure 18 (*right*) that the astigmatism and spherical aberration remain negligible, whilst the coma deviation error is dominating with a 0.4 He-Ne wavelength (λ = 633 nm) in RMS fringes. This is due to a lack of the over-thickness at the tip or mainly a non-uniformity of the collarette radial thickness and/or some decentering of the force actuator. The surface plot shows a local deviation is the central area. However, for a zoom range up to *f*/5 and a 14 mm clear aperture, the RMS results are not far from the limit of diffraction.

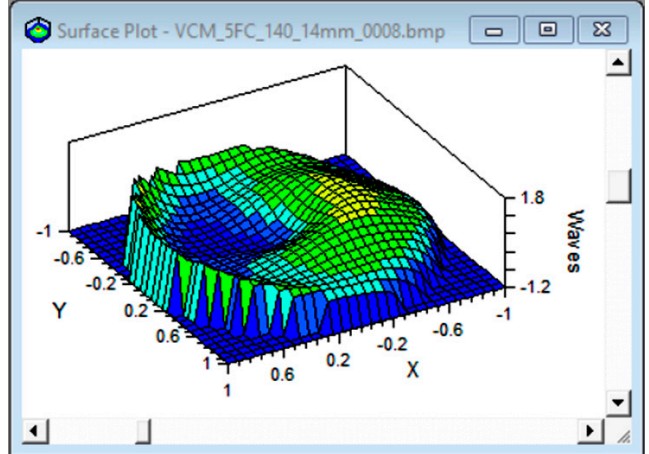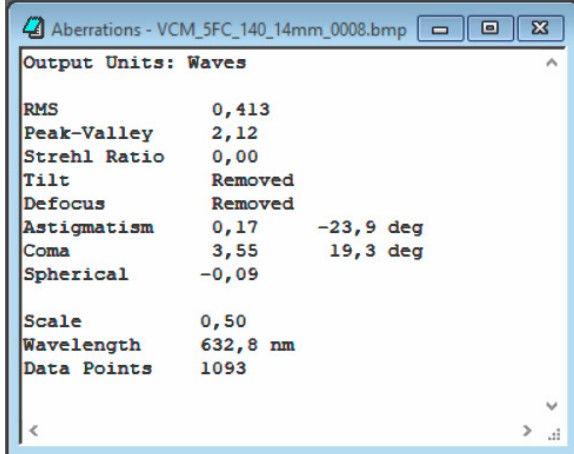

**Figure 18.** Surface plot (**left**) and aberration table (**right**) of VCM#5 at full clear aperture 14 mm for the maximum curvature deformation at $f/5$, corresponding to a curvature radius 140 mm.

## 7. Conclusions

Compared to both cases of the *cycloid-like* VCMs [17] and present *tulip-like* VCMs, this FEA modeling with a plate-collarette-ring in a one-piece assembly will require high geometric precision in the execution. This because, for a similar deflection amplitude up to $f/3.5$, the edge thickness of the plate is much thinner in the *tulip-like* case.

For a central force range where $F \leq 10$ daN, our results show, from the FEA Nastran modeling of a *tulip-like* VCM, that a *diffraction-limited* surface quality is achieved. The flexural deviations to a paraboloid all remain smaller than 0.1 μm of RMS error for a 13 mm optical aperture and a zoom range from $f/\infty$ to $f/3.5$.

Compared to our previous results from the thin single-plate deflection analytic theories, leading us to the construction of the first prototype giving a zoom range restrained to $f/5$, the present modeling results should significantly help in the future construction of such VCMs with a larger zoom range.

## 8. Patents

Lemaitre, G.R. French patent submitted 1976, "Miroirs à focale variable et procédés d'obtention", registered No. FR2343262, 1977.

Lemaitre, G.R. US patent submitted 1976, "Mirrors with a variable focal distance", registered No. US4119366, 1978.

**Author Contributions:** Conceptualization, G.R.L. and P.V.; methodology, P.V. and P.L.; software, P.V. and P.L.; validation, G.R.L., P.V. and P.L.; formal analysis, P.V. and P.L.; investigation, G.R.L.; resources, P.V. and P.L.; data curation, P.V. and P.L.; writing—original draft preparation, G.R.L. and P.V.; writing—review and editing, G.R.L.; visualization, G.R.L., P.V. and P.L.; supervision, G.R.L. All authors have read and agreed to the published version of the manuscript.

**Funding:** This research received no external funding.

**Data Availability Statement:** Not applicable.

**Conflicts of Interest:** The authors declare no conflict of interest.

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
