# Peer review of "Active Optics—Progress in Modeling of Tulip-like Variable Curvature Mirrors"

_optics, doi:10.3390/opt4010004_

Round 1

Reviewer 1 Report

The authors built a tulip-like variable curvature mirror and tried unsuccessfully to support their contribution with simulations and experimental measurements. The paper is not written correctly, making it difficult to review.  For example, 

1) The authors in the abstract explain two types of variable curvature mirrors form but end up showing results for one tulip-like type. 

2) The information provided in the introduction is poor and similar to the abstract; they need to extend this part to assets their contribution correctly. 

I could provide more recommendations, but I think the authors first need to review the document more carefully and explain each part of their submission more appropriately. 

Regards

The reviewer. 

Reviewer 2 Report

Authors showed design of the tulip-like variable curvature mirrors. However, there are no research motivation as shown in abstract section. In addition, there are no previous study without references. English grammar and styles need to be improved so please ask professional English service or ask native English colleague. Therefore, authors extensively revise the manuscript with critical comments as below.

1. There are too small numbers of the references. Authors need to provide the previous research in the introduction section.

2. Authors must provide comparison data of the proposed research with other scholar's work.

3. Authors must use MDPI styles for the manuscript.

4. Reference format is wrong.

5. In conclusion, motivation of the research and summary of the data and future work need to be provided.

6. No mark for each separate Figure 17.

7. Fig, 17 -> Fig. 17.

8. Please use Figure instead of Fig.

9. Title has better show novelty of the proposed research.

Reviewer 3 Report

Dear authors,

Thank you very much for your paper "Active Optics – Progress and Modeling of Tulip-like Variable Curvature Mirrors", which I enjoyed reading. I am pleased with the differentiated approach, which also includes buckling instabilities. With the following comments I would like to help to further improve the paper.

Kind regards!

Please explain the assumptions you made and the conclusions you drew. This way you will help readers who are not as deep in your field as you are to easily follow the content.

Please add a state of the science/ art on adaptive mirrors that takes into account more than just your own preliminary work.

Sometimes, additional illustrations, especially cross-sections of the geometry with dimensions, would greatly help the reader. I discuss this in more detail below.

L19 Here you mention the "non-linear static flexural option". Please use this term in the main part of the paper as well.

L32 Please clarify that the three different geometries result from three different sets of boundary conditions (acting forces).

L34 Please position Figure 1 here, this helps to understand the paper.

L36 Please explain where the bending moment would be applied - gladly with a sketch.

L38 A reference is missing here

L44 What is the "middle surface"? This term appears in several places without being explained.

L47 Please explain the abbreviation LAM.

L64 What does the index 20 stand for?

L65 Is A_20 constant? If yes so is 1/R, which is called variable - please clarify.

L75 Please explain the criterion behind "convenient".

L86 It is not clear here why three thicknesses T_20 are mentioned.

L92 Please explain why only this case is considered.

L93 Please explain briefly why an FEM analysis is required.

L96 Please illustrate the moments M_r and M_t in a sketch.

L140 This is surely not about the change of curvature but about the direction of curvature.

L142 Please add the reference where this was shown.

L143 Please add the criterion for “small”, if possible.

L149 A more suitable metal is mentioned here. Why is this not used? If no scientific answer is possible here, e.g. mention this metal as another possibility for optimization.

L149 What does “linear” refer to?

L160ff Please refer to the modified design with the collarette in section 4.2 and not here, otherwise it will confuse the reader unnecessarily. Accordingly, please position Figure 2 in section 4.2 as well.

L167 Please position 0 as index.

L172 Please show the new design in a cross-section view.

L176 Please explain why a different profile is used for the outer area. Please add the parameters for the transitions etc. to the figure mentioned above.

L185 The nonlinearity cannot be easily using Figure 3. A table with values for the force and delta z(rho=0) and their quotient would help.

L191 Please write t(rho=1) instead of t(1) or specify it in mm. Please use a consistent solution (mm or dimensionless) in figures and caption.

L193 Please do not state the equation in the figure, which is only valid for a part of the profile. This is otherwise misleading.

L194 Please explain (in the text) what was averaged.

L199 Please describe the "important local effect". Does this effect really occur or is it a numerical instability?

L200 Please clarify what is to be optimized here and how.

L204 Compare L193.

L209 What deformation should be enabled by adding the collarette?

L212 Please state parameters.

L218 Please define the flexibility-ratio.

L227 Please adjust the scaling of the vertical axis so that the graphs are fully displayed. Why are there no values for radii close to 8mm?

L234ff Two approaches to optimize the geometry are presented here. Please be sure to explain where these come from and how the values for epsilon were determined.

L238 max(delta z) instead of delta z.

L240 Please explain the term “over-thickness”.

L252 Please explain t and t* here.

L271 Please orient the geometry shown here as e.g. in figure 11.

L279 Here 10daN, in table 2 12daN.

L281 Here 13mm, in L231 14mm.

L287 Define t_0 as mean thickness much further ahead in the paper.

L291/292 Two different zoom values are given here - please explain.

L297 Please refer to figure 13.

L306 Reference required for the modeling of the creeping effect.

L311 Swap order of center and right image.

L312 10.1daN

L315 Which criterion was used to identify the most stressed areas?

L331ff This section refers to the fabrication and measurement of the non-optimized geometry. It is a pity that the optimized one is not also considered accordingly. What is the added value of this section compared to the state of the science?

L357 Please give the HeNe wavelength also as a numerical value.

L357f Please move the discussion of possible influencing factors to the actual text.

L368 Which publication is cited here? Please indicate the appropriate reference.

Reviewer 4 Report

ms:  optics-2078587   Active Optics – Progress and Modeling of Tulip-like Variable Curvature Mirrors

Authors: Gerard R. Lemaitre, Pascal Vola , Patrick Lanzoni

A. Overview

1. In this manuscript the authors report on the theory and simulation of Tulip-like Variable Curvature Mirrors. Also they present the experimental realization and results of a prototype tulip-like VCM.

2. It is written in reasonable English.

3. The authors have acknowledged recent related research.

4. As long as my knowledge, the work presented is original.

5. The contents of the manuscript are not clearly expressed, starting with the organization of the text which must follow the standard style.

B. Detailed analysis.

Abstract – is very lengthy: state what is the issue, what you have done, how did you do it, the results and the novelty.

1. Introduction: provides an interesting approach to the subject and there are up to date references. No theory or equation here.

2. Organize de text following a simple section structure: Theory, Simulation methodology, Simulation Results, Experimental Result (including Materials and Methods)

3. There are too many figures. Are all necessary? Write the Caption in the usual way.

4. Section 7. Patent  -  is there for what?

C. Overall assessment

The work presented here is interesting and has potential for further development in the field.

However, in my opinion the manuscript must be written form the beginning. I advise Major corrections in the manuscript

D. Review Criteria

1. Scope of Journal

Rating: Medium

2. Novelty and Impact

Rating: Medium

3. Technical Content

Rating: Medium

4. Presentation Quality

Rating: LOW

Round 2

Reviewer 1 Report

The modifications helped me to understand the author's works, although I still found some English mistakes, even in the blue-marked text. For example, in line 321, page 321 "force-actiator". 

I strongly recommend using a payable edition service; Grammarly pro could be helpful in this case. 

Regards

The reviewer. 

I recommend publication 

Reviewer 2 Report

Authors answered the questions so the paper can be accepted.

Reviewer 4 Report

Manuscript :  optics-2078587 R1   Active Optics – Progress in Modeling of Tulip-like Variable Curvature Mirrors

Authors: Gerard R. Lemaitre, Pascal Vola , Patrick Lanzoni

A. Overview

In this manuscript the authors report on the theory and simulation of Tulip-like Variable Curvature Mirrors. Also they present the experimental realization and results of a prototype tulip-like VCM.

B. Overall assessment

The work presented here is interesting and has potential for further development in the field.

The authors answered the reviewer’s questions and queries.

In my opinion the work can be accepted for publication given that the authors made changes in the manuscript and improved it

C. Review Criteria

1. Scope of Journal

Rating: Medium

2. Novelty and Impact

Rating: Medium

3. Technical Content

Rating: Medium

4. Presentation Quality

Rating: Medium
